# Effect of Different Nutritional Supplements on Glucose Response of Complete Meals in Two Crossover Studies

**DOI:** 10.3390/nu14132674

**Published:** 2022-06-28

**Authors:** Nele Gheldof, Celia Francey, Andreas Rytz, Léonie Egli, Frederik Delodder, Lionel Bovetto, Nathalie Piccardi, Christian Darimont

**Affiliations:** 1Nestlé Research, Institute of Health Sciences, CH-1000 Lausanne, Switzerland; nele.gheldof@yahoo.com (N.G.); celia.francey@ik.me (C.F.); leonie.egli@gmail.com (L.E.); 2Nestlé Research, Clinical Research Unit, CH-1000 Lausanne, Switzerland; andreas.rytz@rdls.nestle.com (A.R.); frederik.delodder@rdls.nestle.com (F.D.); npiccardi@gmail.com (N.P.); 3Nestlé Research, Institute of Material Science, CH-1000 Lausanne, Switzerland; lionel.bovetto@rdls.nestle.com

**Keywords:** blood glucose, whey protein, mulberry (*Morus alba*) leaf extract, glycemic response, diabetes mellitus

## Abstract

Postprandial hyperglycemia is an important risk factor in the development and progression of type-2 diabetes and cardiometabolic diseases. Therefore, maintaining a low postprandial glucose response is key in preventing these diseases. Carbohydrate-rich meals are the main drivers of excessive glycemic excursions during the day. The consumption of whey protein premeals or mulberry leaf extract was reported to reduce postprandial glycemia through different mechanisms of action. The efficacy of these interventions was shown to be affected by the timing of the consumption or product characteristics. Two randomised crossover studies were performed, aiming to identify the optimal conditions to improve the efficacy of these nutritional supplements in reducing a glycemic response. The acute postprandial glycemic response was monitored with a continuous glucose monitoring device. The first study revealed that a preparation featuring 10 g of whey protein microgel reduced the postprandial glucose response by up to 30% (*p* = 0.001) and was more efficient than the whey protein isolates, independently of whether the preparation was ingested 30 or 10 min before a complete 320 kcal breakfast. The second study revealed that a preparation featuring 250 mg mulberry leaf extract was more efficient if it was taken together with a complete 510 kcal meal (−34%, *p* < 0.001) rather than ingested 5 min before (−26%, *p* = 0.002). These findings demonstrate that the efficacy of whey proteins premeal and mulberry leaf extracts can be optimised to provide potential nutritional solutions to lower the risk of type-2 diabetes or its complications.

## 1. Introduction

Controlling postprandial glucose response (PPGR) is important in both the management and prevention of type-2 diabetes (T2D) [1,2]. PPGR was shown to be the main contributor to the total glucose fluctuations in T2D or patients with prediabetes, in whom haemoglobin A1c levels need to be maintained below 8% for optimal glucose control [3]. Controlling PPGR in the overweight and obese population, also at risk for T2D, appears to be the key to preventing this disease. However, as prediabetes is rapidly increasing worldwide and is not only associated with body mass index (BMI) but also with age, prevention should also start with healthy and lean people [4].

The macronutrient composition of a meal, and especially its quantity and quality of carbohydrates (CHO), are the main drivers of PPGR rise. Yet, nutritional supplements have been reported to lower the PPGR of CHO-rich meals independently of a change in their macronutrient content. These supplements can be taken either before or during a meal, depending on their compositions and mechanisms of action.

One of the most documented ingredients for mediating such an effect is whey protein, which can reduce the glucose response of a meal in healthy or T2D subjects when taken within a maximum of 30 min before a meal [5,6,7]. The studies that were conducted on lean and healthy subjects have shown that in contrast to a high dose of whey protein (50 g), 10 g of whey protein taken 30 min before a meal can lower glucose excursions by delaying gastric emptying without stimulating insulin secretion [5,8]. Such a mechanism of action of whey protein on glycemic PPGR seems to be specific to a premeal administration. The co-ingestion of whey protein with a meal was reported to lower PPGR, essentially by increasing insulin secretion through higher doses of protein [9]. Only a few studies have compared the potential impact of the timing of whey protein consumption or of the different forms of whey protein on its effectiveness in lowering PPGR. No difference in PPGR was observed when subjects with metabolic syndrome took 17.6 g of intact whey protein, either 30 or 15 min before a fat-rich meal. However, gastric emptying was more pronounced when the protein was taken 15 min before [10]. In contrast to the intact whey protein, 10 g of hydrolysed whey protein premeal, having a faster rate of amino acid absorption, did not reduce PPGR. These results suggest that a slower amino acid absorption may favour whey protein premeal efficacy on PPGR [8,11]. The whey protein microgels (WPM) containing protein aggregates were reported to delay amino acid absorption as compared to the intact whey protein [12]. It was, therefore, interesting to determine if a premeal with whey protein aggregates would affect the glucose response of a meal differently than the intact whey protein isolates (WPI), and if such effects could also be observed in overweight subjects with a higher risk of impaired glucose tolerance.

Another approach for lowering PPGR is by inhibiting glucose absorption. Amongst the different ingredients and plant extracts described for this effect, mulberry leaf extract (MLE) has been consistently reported in studies for lowering an acute blood glucose response upon CHO ingestion, which was confirmed in a meta-analysis [13]. The main active ingredient in MLE is 1-deoxynojirimycin (DNJ), which acts through potent reversible, competitive α-glucosidase inhibition, and glycogen phosphorylase inhibition, as evidenced by in vitro studies [14,15]. Studies showed that MLE, at dosages ranging from 6 to 36 mg DNJ, was efficient in lowering PPGR in healthy subjects [16,17,18,19], as well as in people with an impaired glucose metabolism [20], or T2D [21,22,23]. In most studies, the MLE efficacy was demonstrated when the extract was ingested as a capsule prior to a pure CHO (starch, maltodextrin, sucrose) ingestion [16,17,18,19,20] or consumption of a simple meal (pure rice, porridge, or cornflakes) [14,20]. To our knowledge, the efficacy of such a competitive glucosidase inhibitor has not been compared when taken before or mixed within a complex, balanced meal, rich in CHO and with non-negligible amounts of lipids and proteins.

The objective of this research was to determine if the effectiveness of these two nutritional approaches reported lowering PPGR and if it could be improved by the timing of consumption or, for the whey protein, by a different protein structure affecting amino acid absorption. To address this, we designed two independent dietary interventions with diverse meal compositions and timing regimens: the first one used a whey protein premeal, and the second study was with the MLE at the lowest effective doses already reported in the literature to avoid any potential impact on tolerance [8,18]. The two studies were performed on healthy subjects. In the first study, the effects of 10 g of WPI were compared with the WPM in overweight volunteers. The two protein forms were tested either 10 or 30 min before a standard breakfast. In the second study, the effect of 250 mg of MLE was tested in lean subjects when taken just before or during a complete meal.

## 2. Materials and Methods

### 2.1. Samples and Interventions

Nine interventions were tested in two studies. These interventions systematically varied the active compound of the supplement and the timing of administration (Table 1). Three supplements were tested vs. the control, pure water. The first supplement (WPI) was a drink composed of a whey protein preparation (Whey Basics, Pure Encapsulation, Sudbury, MA, USA) reconstituted in 100 mL of water. The second supplement (WPM) was a drink composed of 100 mL of a WPM solution, produced from a native whey protein isolate (Pronativ95 from Lactalis Ingredients, Bourgbarré, France), as previously described [24]. For this study, the concentration step was done by conventional evaporation. The WPI and WPM test products contained 78.9% and 86.8% of whey protein and 8.4% and 10.7% of caseins, respectively. The third supplement (MLE) was 250 mg of mulberry (*Morus alba*) leaf extract (5% Reducose^®^, Phynova, Witney, UK), containing 12.5 mg of DNJ that was consumed either before or during the meal. When consumed before, it was reconstituted in 200 mL of water; if consumed during the meal, it was mixed in the standardised rice meal in order to be consumed over the entire meal, at the same time as CHO. The control intervention for the MLE study consisted of 200 mL of water taken before the meal.

In the protein premeal study, the standardised meal was a breakfast composed of 56 g of white bread (2 slices), 25 g of jam and a glass of 330 mL of orange juice. In the MLE study, the standardised meal was composed of 150 g of boiled white jasmine rice, 25 g of white bread, 80 g of curry sauce and 80 g of chicken breast slices. The macronutrient composition of these two meals, as well as the estimated glycemic load [25], which was 48 g of a glucose equivalent in both studies, are described in Table 2.

### 2.2. Design of the Studies

For the protein premeal study, 15 healthy subjects (9 women, 6 men) were recruited with a mean ± SD age = 49 ± 8 years (inclusion criteria: 40–65 y, a BMI = 31.2 ± 2.8 kg/m^2^ (inclusion criteria: BMI > 27 kg/m^2^, with a sedentary lifestyle, not exceeding 30 min of walking per day) and fasting glucose = 5.4 ± 0.6 mmol/L. The sample size was deduced from a previous study that included 10 healthy young men and showed a significant effect of a 10 g whey protein premeal on the PPGR of a standard meal [7]. Assuming a similar effect size, but with an increased variability due to an increased BMI of the subjects, the sample size was set to *n* = 15.

For the MLE study, 30 subjects (11 women, 19 men) were recruited with a mean ± SD age = 31 ± 7 years (inclusion criteria: 18–45 y), a BMI = 22.9 ± 2.2 kg/m^2^ (inclusion criteria: BMI between 20 and 29.9 kg/m^2^) and fasting glucose = 5.0 ± 0.5 mmol/L. The sample size was deduced from two previous studies that both reported a 25% reduction in PPGR of either a rice-based standard meal or a load of 50 g of maltodextrin [16,18]. Assuming similar effect sizes and variabilities, the calculated effect size was *n* = 30 to reach a power of 80%.

The key exclusion criteria were the same in the two studies, namely any metabolic disease, including diabetes or chronic drug intake, a known allergy and intolerance to components of the test products, smoking, and contraindications to the sensor’s placement (e.g., skin hypersensitivity).

The day before each testing visit—with one test condition per visit—the subjects were required to refrain from consuming alcohol and performing strenuous exercise. They were asked to come to the Nestlé Research Center at 8 h 00, after a 12 h fasting, without taking any medication, such as aspirin or supplements containing vitamin C that may affect continuous glucose monitoring (CGM). 

The glucose response was measured with a CGM device (FreeStyle Libre^®^, Abbott, Chicago, IL, USA), measuring the interstitial glucose concentration every 15 min [26,27,28,29]. The sensor was placed on the non-dominant arm of each subject at least 24 h before the first visit, and a reader, as well as the instructions for its use, were provided. If a sensor was lost during the study, it was replaced, and the subject could resume the study with the next testing visit at least 24 h after the sensor’s insertion. The sensor was removed at the end of the study by a clinical staff member.

Both studies were monocentric, with a crossover, randomised and open design. The subjects were randomly assigned to a sequence of a Williams Latin square that balanced the position and carry-over effect to minimise potential bias [30]. Since the subjects could test all experimental conditions once using the same CGM sensor, randomisation could be performed without any restrictions, such as blocking (see flowchart in Figure 1).

The subjects signed an informed consent form as per local regulations, and the study protocols were reviewed and approved by the Ethics Committee of Canton de Vaud (Lausanne, Switzerland, CER-VD 2019-01814, CER-VD 2018-00934) and registered at clinicaltrials.gov (NCT05112133, NCT05112146).

### 2.3. Data Analyses

The primary endpoint in these studies was the 2 hPPGR (2h-PPGR) incremental area under the curve (iAUC) that was calculated using the trapezoid method for each individual PPGR after the standardised meal. The additional endpoints of interest were the maximal incremental glucose value (iCmax) and the time to reach this value (Tmax). At the beginning of each visit, the subjects scanned the sensor with the reader right before the standardised meal intake (T0). The descriptive statistics (Mean, SEM) were tabulated and visualised. The means were compared using paired t-tests with a two-sided significance level set at 5%, following the established standards [31]. A sensitivity analysis was performed by using a mixed model to impute the possible missing data and to consider the potential systematic position or carry-over effects [32]. Since none of these effects were close to reaching statistical significance, this analysis is not further presented. 

## 3. Results

### 3.1. Average 2h-PPGR Curves

The average 2h-PPGR curves show that for the standardised meals with comparable estimated glucose loads, eGL = 48, the protein premeal study led to higher PPGR peaks than the MLE study. The protein premeal study that included older (49 vs. 31 y) and more overweight subjects (BMI = 31.2 vs. 22.9 kg/m^2^) also showed a higher average glucose baseline (5.4 vs. 5.0 mmol/L). In both studies, the average curve was not back to baseline after 2 h. 

The mean PPGR curves of the different interventions are visualised together with the standard error (Figure 2). These mean curves were established for the *n* = 14 completers of the protein premeal study and all 30 subjects enrolled in the MLE study. All subsequent analyses were performed on these analysis sets. One subject could not complete the protein premeal study because he always lost the sensor during the first 24 h after placement.

### 3.2. Average 2h-iAUC, iCmax and Tmax

The average 2h-iAUC, iCmax and Tmax (mean ± SE) are tabulated for the nine tested conditions (Table 3).

For the pairwise comparisons of the highest interest, the relative differences in each iAUC, as well as the absolute differences in iCmax and Tmax, are tabulated together with the corresponding *p*-values (Table 4).

It is shown that compared to Control 30, WPM30 significantly decreased 2h-iAUC (−30%, *p* = 0.001), while WPI30 only reached a trend (−14%, *p* = 0.104). The 2h-iAUC of WPM30 was furthermore significantly lower than WPI30 (−19%, *p* = 0.042). In terms of the iCmax, the effect was significant for both WPM30 (−1.09 mmol/L, *p* = 0.001) and WPI30 (−0.70 mmol/L, *p* = 0.019) vs. Control 30. The Tmax was further delayed by 9 min, from 50 to 59 min for both premeals, but this effect was not statistically significant.

In the frame of the same protein premeal study, it is shown that compared to Control 10, WPM10 significantly decreased 2h-iAUC (−25%, *p* = 0.019), while WPI10 only reached a trend (−18%, *p* = 0.077). The 2h-iAUC of WPM10 was not significantly lower than WPI10 (−9%, *p* = 0.375). In terms of the iCmax, the effect was significant for both WPM10 (−1.13 mmol/L, *p* = 0.004) and WPI10 (−0.94 mmol/L, *p* = 0.009) vs. Control 10, while the Tmax was not significantly impacted.

Although the observed effects were larger when the administration was 30 min before rather than 10 min before the standardised breakfast, the direct comparison of the two administration modes was not significantly different with neither the WPM nor WPI for 2h-iAUC nor the iCmax. However, the interstitial glucose responses after the WPM or WPI, taken 30 min before a meal, reached their Tmax significantly later than when they were taken 10 min before (WPI: +14 ± 6 min, *p* = 0.042; WPM: +13 ± 5 min, *p* = 0.033). 

In the frame of the MLE study, it is shown that when compared to the control, 2h-iAUC was significantly reduced by the MLE Before (−26%, *p* = 0.002) and MLE During (−34%, *p* < 0.001) the standardised meal. The 2h-iAUC of the MLE During was furthermore significantly lower than the MLE Before (−10%, *p* = 0.050). In terms of iCmax, the effect was significant for both the MLE Before (−0.68 mmol/L, *p* = 0.001) and MLE During (−0.84 mmol/L, *p* < 0.001). The Tmax was further delayed by 22–25 min, from 60 to 82–85 min for both premeals; however, this effect appeared as significant only for the MLE Before (*p* = 0.023).

## 4. Discussion

The presented studies tested if the efficacy of whey protein premeal and MLE, two nutritional supplements previously reported to lower PPGR, could be optimised by changing the timing of their consumption or protein structure. 

The protein premeal study showed that the consumption time (10 or 30 min before the meal) did not have any significant impact on the effects of the WPI or WPM premeals on the glucose response of the subsequent meal (iAUC, iCmax, Tmax). These results are consistent with previous results showing that consuming 17.6 g WPI 15 or 30 min before a fat-rich meal did not differentially alter PPGR in subjects with metabolic syndrome [10]. The reduction in the postprandial interstitial glucose observed in our study was similar to the effect observed in the blood glucose response after consuming 10 g of WPI taken 30 min before eating a pizza (about −30% in iCmax; [8]). This suggests that the measurement of interstitial glucose by a CGM device can be used as a good and less invasive alternative to blood sampling. In addition, our study confirms that the effect of 10 g of WPI previously observed in lean subjects [5,8] was also observed in overweight volunteers. Interestingly, the WPM induced a greater reduction in the iAUC and Cmax than the WPI-preload at both consumption times and, more importantly, when taken 30 min before. The mechanism of action explaining this stronger PPGR reduction induced by the WPM premeal is unclear. Although the WPM solution had a slightly higher whey protein content than the WPI preparation (+0.79 g in the 10 g of total protein WPM solution), it is unlikely that it could explain the improved PPGR reduction in the WPM treatment. The low dose of whey protein premeal (10 g) was reported to reduce PPGR through a decrease in the gastric emptying rate [13]. In addition, when hydrolysed, 10 g of whey protein premeal lost its capacity to reduce PPGR, suggesting that a faster amino acid absorption could impair the effectiveness of the whey protein premeal on the glucose response of the following meal. Because WPM was shown to have delayed protein digestion as compared to WPI [12], it can be speculated that a lower amino acid absorption might favour a PPGR reduction. The precise mechanism of action of WPM vs. WPI needs to be further explored. 

In the second study evaluating the PPGR effects of MLE, we confirmed that MLE could decrease the PPGR of a complete meal. In a previous study, the same dose of 12.5 mg DNJ, in a capsule, in co-ingestion with maltodextrin, resulted in a 14% reduction of 2h-iAUC [18]. Another study reported a 24% reduction of 2h-iAUC when a smaller dose of 8 mg DNJ was taken before porridge [33]. The effect observed in the present study, when MLE was absorbed 5 min before the meal, was of a similar magnitude (−26%). Interestingly, we demonstrated that the timing of administration is an important aspect in obtaining the optimal effects of MLE on PPGR. Indeed, MLE induced a stronger reduction in the glucose response when mixed with the meal. It is reasonable to expect that a maximal effect will be observed when the DNJ reaches the small intestine at the same time as CHO in the food to compete for binding to the α-glucosidase enzymes.

Such interventions with the MLE or WPM premeal appear as potential convenient solutions for subjects with impaired glucose tolerance or diabetes for their daily blood glucose control. However, even if the MLE and WPI premeal were reported to significantly improve glucose management when taken for several consecutive days [20,34], it remains to be demonstrated that these optimal interventions would mediate superior efficacy when taken chronically. The two performed studies have some limitations. As a first limitation, the impact of the interventions on other metabolic markers, such as blood lipids and insulin secretion, was not assessed in the two studies. Previous studies testing 10 g of WPI premeal or the MLE extracts have all shown that a reduction in PPGR was associated with a decrease in insulin secretion. Therefore, it is tempting to speculate that the PPGR reduction observed in these studies might also be associated with lower postprandial insulin responses. The second limitation is that the two studies were performed on healthy subjects, and the relevance of these findings in patients with diabetes needs to be further confirmed. However, both the WPI premeal and MLE were shown to reduce PPGR in both healthy subjects and patients with diabetes [11,22]. Therefore, it is highly likely that our observations with the WPM premeal and MLE will be relevant for subjects with diabetes, as well. 

## 5. Conclusions

The management of postprandial glycaemia is a key concern for people with prediabetes or diabetes. In this research, we demonstrated that the efficacy of nutritional solutions, such as protein premeal or MLE, to reduce glucose excursion of a complete meal could be improved by either the timing of consumption or by a whey protein structure. 

Although the two presented studies were performed independently of each other, with different interventions, different standardised meals, and different inclusion criteria for the subjects, it is remarkable to notice that the reduction of 2h-iAUC is very large and of similar magnitudes, reaching maximally 30–34% for meals with estimated glycemic loads close to 50 g.

These findings not only further advance the technical applications of these nutritional supplements in different food formats, but also the relevance of obtaining optimal health benefits to lower the risk of T2D or its complications.

## Figures and Tables

**Figure 1 nutrients-14-02674-f001:**
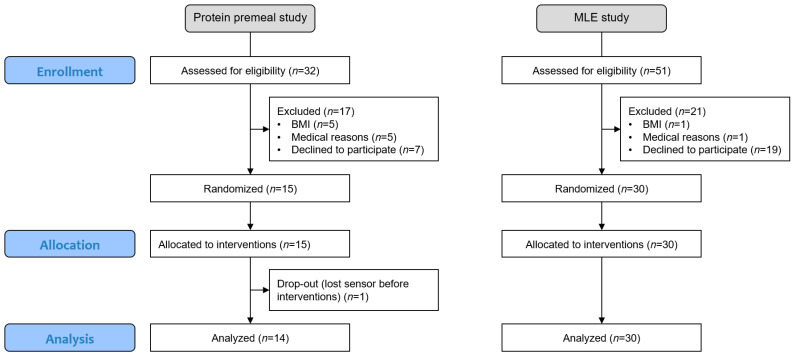
Flowchart of the two studies, including enrollment, allocation, and analysis.

**Figure 2 nutrients-14-02674-f002:**
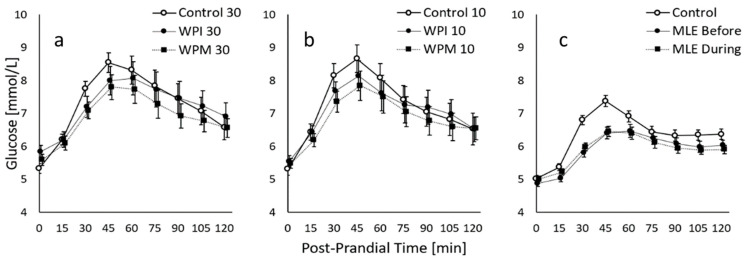
Average 2h-PPGR of the protein premeal study featuring *n* = 14 completers, with premeal taken either 30 min before (**a**) or 10 min before (**b**) the complete breakfast with an estimated GL of 48 g, and average 2h-PPGR of the MLE study featuring *n* = 30 subjects (**c**). The figure shows cross-sectional values with mean ± SE.

**Table 1 nutrients-14-02674-t001:** Nine interventions tested in two studies, featuring three active ingredients (WPI: Whey Protein Isolates, WPM: Whey Protein Microgel, and MLE: Mulberry Leaf Extract).

Study	Intervention	Supplement [g Active + mL Water]	Timing [min before Meal]
Protein Premeal	Control 30	0 g + 100 mL	30
	WPI 30	10 g WPI + 100 mL	30
	WPM 30	10 g WPM + 100 mL	30
	Control 10	0 g + 100 mL	10
	WPI 10	10 g WPI + 100 mL	10
	WPM 10	10 g WPM + 100 mL	10
MLE	Control	0 mg + 200 mL	5
	MLE Before	250 mg MLE + 200 mL	5
	MLE During	250 mg MLE + 200 mL	0

**Table 2 nutrients-14-02674-t002:** Macronutrient composition and estimated glycemic load (eGL) of the standardised meals served in the two studies (CHO: Carbohydrates).

	Protein Premeal Study	MLE Study
Energy [kcal]	320	510
CHO [g (%kcal)]	71.0 (89%)	72.5 (57%)
Sugars [g (%kcal)]	43.5 (54%)	4.5 (4%)
Protein [g (%kcal)]	5.0 (6%)	24.9 (19%)
Fat [g (%kcal)]	1.8 (5%)	13.4 (24%)
eGL [g]	48	48

**Table 3 nutrients-14-02674-t003:** Descriptive statistics (mean ± SE) for the nine tested conditions with *n* = 14 for the protein premeal study and *n* = 30 for the MLE study.

	2h-iAUC [mmol/L × min]	iCmax [mmol/L]	Tmax [min]
Control 30	245 ± 30	3.50 ± 0.33	50 ± 5
WPI 30	212 ± 30	2.80 ± 0.33	59 ± 5
WPM 30	172 ± 26	2.41 ± 0.33	59 ± 5
Control 10	247 ± 29	3.77 ± 0.39	51 ± 6
WPI 10	203 ± 32	2.83 ± 0.36	45 ± 4
WPM 10	185 ± 28	2.65 ± 0.32	46 ± 5
Control	167 ± 12	2.45 ± 0.14	60 ± 7
MLE before	123 ± 12	1.77 ± 0.13	85 ± 9
MLE during	111 ± 10	1.61 ± 0.12	82 ± 9

**Table 4 nutrients-14-02674-t004:** Pairwise comparison with effect size (mean ± SE) and *p*-value (paired *t*-test) for the nine comparisons of the highest interest, with *n* = 14 for the protein premeal study and *n* = 30 for the MLE study.

	2h-iAUC [%]	iCmax [mmol/L]	Tmax [min]
WPI 30—Control 30	−14 ± 8 (*p* = 0.104)	−0.70 ± 0.26 (*p* = 0.019)	9 ± 5 (*p* = 0.104)
WPM 30—Control 30	−30 ± 7 (*p* = 0.001)	−1.09 ± 0.24 (*p* = 0.001)	9 ± 7 (*p* = 0.218)
WPM30—WPI30	−19 ± 8 (*p* = 0.042)	−0.40 ± 0.22 (*p* = 0.100)	0 ± 8 (*p* = 1.000)
WPI 10—Control 10	−18 ± 9 (*p* = 0.077)	−0.94 ± 0.31 (*p* = 0.009)	−6 ± 6 (*p* = 0.290)
WPM 10—Control 10	−25 ± 9 (*p* = 0.019)	−1.13 ± 0.33 (*p* = 0.004)	−5 ± 7 (*p* = 0.444)
WPM10—WPI10	−9 ± 10 (*p* = 0.375)	−0.19 ± 0.29 (*p* = 0.534)	1 ± 7 (*p* = 0.876)
MLE before—Control	−26 ± 7 (*p* = 0.002)	−0.68 ± 0.17 (*p* = 0.001)	25 ± 9 (*p* = 0.023)
MLE during—Control	−34 ± 7 (*p* < 0.001)	−0.84 ± 0.15 (*p* < 0.001)	22 ± 11 (*p* = 0.206)
MLE during—MLE before	−10 ± 7 (*p* = 0.050)	−0.16 ± 0.12 (*p* = 0.046)	−3 ± 10 (*p* = 0.420)

## Data Availability

The data presented in this study are available on request from the corresponding author. The data are not publicly available due to intellectual property rights.

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
