# Peer review of "Effect of Different Nutritional Supplements on Glucose Response of Complete Meals in Two Crossover Studies"

_nutrients, 2022, doi:10.3390/nu14132674_

Round 1

Reviewer 1 Report

In the paper, nutrients-1764419 authors presented two compositions of compounds that were tested in terms of PPGR, that act by a different mechanism.

The overall paper looks nice and obtained results are interesting and valuable, especially in terms of the time of administration of individual components. In order to improve the work, to make it more readable for the reader, I propose to introduce the following changes.

1.      Please give the reason why WPM and WPI were tested before breakfast and MLE as a standard meal.

2.      On what basis the concentrations of the compounds used were determined.

3.      Give the reason why the criteria of the qualification of participants differed between the WPM and the MLE

4.      Exactly how many participants were contained in each group presented in Table 1. How many individuals received WPI 30, how many WPM 30, etc.

5.      Why there was no control for MLE during. As I understand the control refers to MLE before, is that correct?

6.      Please systematize the units provided in the manuscript, mainly in terms of giving with or without spaces

Author Response

Reviewer #1 

  1. Please give the reason why WPM and WPI were tested before breakfast and MLE as a standard meal.

In this study the efficacy of different forms of 10g of whey proteins (WPI or WPM) were tested as a pre-meal (10 or 30min before) and not during a meal. The main reason of this choice was because we wanted to use a minimal effective dose of whey protein to avoid any significant impact on daily protein intake and on insulin secretion. Administration of 10 g of whey protein isolate 30 min before a meal has been indeed reported to lower glycemic response of the following meal essentially by delaying gastric emptying and not by stimulating insulin secretion (8).  Since the effect on glycemic response when whey protein is co-ingested with was shown to be mediated by high dose of protein stimulating insulin secretion (9), it didn’t make sense to test such low dose of protein during a meal.

Due to the different mechanism of action of MLE, inhibiting glucosidase, it was taken during the meal when carbohydrate digestion occurs.

The rational for testing whey proteins before a meal is now specified in the Introduction (page 2, line 50-53)

  1. On what basis the concentrations of the compounds used were determined.

The concentrations of the whey protein and MLE that were tested in the studies were chosen based on dose response efficacy data (8, 17) by choosing the minimal effective doses to avoid any tolerance and sensory issues.

This information was added in the Introduction (page 2, lines 85-86).

  1. Give the reason why the criteria of the qualification of participants differed between the WPM and the MLE.

The effect of MLE was tested in lean subjects whereas the effects of 10 g whey protein pre-meal was tested in overweight subjects. As described in the Introduction, the effects of MLE on postprandial glucose were demonstrated in healthy as well as in overweight and diabetic subjects. In contrast, the positive effect of 10g whey protein pre-meal was shown only in healthy and lean subjects (8). Our study confirmed that this effect could also be observed in an overweight population with potential alteration on glucose metabolism.

This clarification was added in the Introduction and Discussion (page 2, lines 48, 65-66 and page 8, lines 248-250).

  1. Exactly how many participants were contained in each group presented in Table 1. How many individuals received WPI 30, how many WPM 30, etc.

We fully agree that this is very important information. Since table 1 only characterizes the nine interventions without relating them to the design – and more specifically to the sample size - of the clinical trial, we preferred to add this information in forms of study flow-charts (new figure 1) as requested by reviewer #2.

  1. Why there was no control for MLE during. As I understand the control refers to MLE before, is that correct?

          Intervention with MLE before and during a meal were indeed compared to an intervention with 200 ml water before the standard meal. A control group consisting in taking 200 ml water during the meal was not considered for this study. Indeed, based on the literature this volume of water was estimated for not reducing glucose response per se. In healthy subjects consumption of 300 ml water was reported to increase glucose response (I. Torsdottir and H. Andersson; Effect on the postprandial glycaemic level of the addition of water to a meal ingested by healthy subjects and Type 2 (non-insulin-dependent) diabetic patients, Diabetologia, 1989). The reduction of glucose response  observed in our study when MLE is administrated in 200ml during a meal cannot therefore be attributed to water.

This specification was added in the Material & Methods (page 3, lines 107-108)

  1. Please systematize the units provided in the manuscript, mainly in terms of giving with or without spaces

Units are now standardized in the manuscript

Reviewer 2 Report

Effect of different nutritional supplements on glucose response 2 of complete meals in two crossover studies

Congratulations on conducting such an interesting study. The authors aim to determine if the effectiveness of WPM premeal and MLE approaches reported to lower PPGR could be improved by the timing of consumption, or, for the whey protein, by a different protein structure affecting amino acid absorption. While the author successfully has reached the conclusion of their study.

The following are my comments and suggestions:

  1. The abstract was written very well and was effective for determining changes in parameters. However, the conclusion part doesn’t match with the rest of the manuscript’s outcomes.
  2. The whole study was about pre-diabetic conditions and none of the diabetic patients was used here, hence, the introduction part should be expanded so that the reader can see the problem, history, and hypothesis regarding prediabetic conditions.  

Methods and Results:

1. What was the rationale for using MLE as a sprinkle over the food, if consumed during the meal, MLE could be reconstituted in 200ml water in this case as well?

2. Why weren't Insulin, Ketone, Cholesterol, HDL-C and LDL-c levels measured in this study? Is there a compelling reason?

3. Please provide the information about patient recruitments and their inclusive and exclusive criteria in flow diagram format as a link to a validated procedure.

4. Kindly provide fasting blood sugar, postprandial, and random blood sugar data in tabular form and show p-values after performing statistical analysis.

Discussion:

Kindy discusses the time frame till when these effects can sustain. What is the best time to use this treatment? 

 So, based on the pre-diabetic study’s conclusion, how do we proceed with managing weight-gain-linked diabetes to alter hyperglycemia conditions?

 References: Journal’s titles of cited articles are not consistent throughout the documents. Some are in the full format while some are in short form. Please make corrections where it is required. 

Author Response

Reviewer #2

  1. The abstract was written very well and was effective for determining changes in parameters. However, the conclusion part doesn’t match with the rest of the manuscript’s outcomes.

Conclusion of the abstract was changed to fit to conclusion of the manuscript (page 1, lines 26-28)

  1. The whole study was about pre-diabetic conditions and none of the diabetic patients was used here, hence, the introduction part should be expanded so that the reader can see the problem, history, and hypothesis regarding prediabetic conditions

The two studies were performed in healthy subjects. Volunteers were not selected based on their prediabetes state (impaired glucose tolerance or impaired fasting glycemia). The choice of healthy volunteers is now more clearly specified in the Introduction (page 2, lines 92-93) as well as the rational for the choice of overweight subjects in the MLE study (page 2, lines 52, 69-70, 94).

Methods and Results:

  1. What was the rationale for using MLE as a sprinkle over the food, if consumed during the meal, MLE could be reconstituted in 200ml water in this case as well?

MLE was mixed with the standardized rice meal to be sure to standardize its consumption between volunteers. In addition, it was important to spread MLE intake over the meal at the same time than carbohydrate consumption and not as a bolus as it could be done with 200ml water.

This information is now added in the Material and Methods (page 3, lines 107-108)

  1. Why weren't Insulin, Ketone, Cholesterol, HDL-C and LDL-c levels measured in this study? Is there a compelling reason?

The primary objectives of these studies were to test the effects of these ingredients on postprandial glucose. For this purpose, continuous glucose monitoring devices were used to measure every 15 min glucose in the interstitial fluid. Blood was not collected; therefore, no other blood metabolic markers could be assessed.

The absence of plasma insulin data was highlighted as a limitation of this study (page 8, lines 278-282) in the initial document. Precision on the lack of other metabolic markers were added in the revised manuscript (page 8, line 284)

  1. Please provide the information about patient recruitments and their inclusive and exclusive criteria in flow diagram format as a link to a validated procedure.

As suggested, a flow-chart of the two studies was added as a new Figure 1           (page 4).

  1. Kindly provide fasting blood sugar, postprandial, and random blood sugar data in tabular form and show p-values after performing statistical analysis.

As mentioned in the question 2, glucose was measured in interstitial fluid with CGM devices and not in blood. All the data on the postprandial glucose reflected by the incremental 2h area under the curve and the Cmax and their statistical analysis are presented in tables 3 and 4 (pages 6 and 7). Fasting glucose level measured at baseline are indicated in the Result section (page 6, line 185).

Discussion:

- Kindy discusses the time frame till when these effects can sustain. What is the best time to use this treatment?

A paragraph was added in the Discussion (page 8, lines 277-282) to discuss this key point.

- So, based on the pre-diabetic study’s conclusion, how do we proceed with managing weight-gain-linked diabetes to alter hyperglycemia conditions?

 The potential positive effects of the interventions on weight management due to either inhibition of glucose absorption for MLE and increased protein intake for WPM are not discussed in the manuscript due to the lack of consistent existing evidence. Chronic treatment with MLE in patients with diabetes did not show any significant effect compared to placebo (20). We would prefer to limit the discussion on the effects of the intervention on glucose management.

- References: Journal’s titles of cited articles are not consistent throughout the documents. Some are in the full format while some are in short form. Please make corrections where it is required. 

Thank you very much for pointing towards this inconsistency. All journal names are now given with full format.